# High-Fat Diet Related Lung Fibrosis-Epigenetic Regulation Matters

**DOI:** 10.3390/biom13030558

**Published:** 2023-03-18

**Authors:** Juntang Yang, Chenxi Liang, Lulu Liu, Lan Wang, Guoying Yu

**Affiliations:** 1State Key Laboratory of Cell Differentiation and Regulation, College of Life Science, Henan Normal University, Xinxiang 453007, China; 2Henan International Joint Laboratory of Pulmonary Fibrosis, Henan Center for Outstanding Overseas Scientists of Pulmonary Fibrosis, Henan Normal University, Xinxiang 453007, China

**Keywords:** high fat diet, lipid biomolecule, epigenetic regulation, pulmonary fibrosis

## Abstract

Pulmonary fibrosis (PF) is an interstitial lung disease characterized by the destruction of the pulmonary parenchyma caused by excessive extracellular matrix deposition. Despite the well-known etiological factors such as senescence, aberrant epithelial cell and fibroblast activation, and chronic inflammation, PF has recently been recognized as a metabolic disease and abnormal lipid signature was observed both in serum and bronchoalveolar lavage fluid (BALF) of PF patients and mice PF model. Clinically, observational studies suggest a significant link between high-fat diet (HFD) and PF as manifested by high intake of saturated fatty acids (SFAs) and meat increases the risk of PF and mice lung fibrosis. However, the possible mechanisms between HFD and PF remain unclear. In the current review we emphasize the diversity effects of the epigenetic dysregulation induced by HFD on the fibrotic factors such as epithelial cell injury, abnormal fibroblast activation and chronic inflammation. Finally, we discuss the potential ways for patients to improve their conditions and emphasize the prospect of targeted therapy based on epigenetic regulation for scientific researchers or drug developers.

## 1. Introduction

Pulmonary fibrosis (PF) is an interstitial lung disease characterized by inflammation and destruction of lung parenchyma caused by accelerated extra cellular matrix (ECM) deposition [1,2] which impaired gas exchange thus decreased quality of life [3]. Despite the well-known etiological factors such as senescence and aging, PF is recently recognized as a metabolic disease [4,5] and abnormal lipid signature has been observed both in serum and bronchoalveolar lavage fluid (BALF) of PF patients and mice model suggesting disturbed lipid metabolism [6,7,8]. Currently there is no cure for PF except for lung transplantation therefore revealing the potential pathogenic factors and possible mechanisms would contribute to the prevention and treatment of this deadly disease.

A high-fat diet (HFD) is well accepted as a critical factor for obesity, a major global health issue [9,10,11] and is associated with poor outcomes in respiratory disease such as acute respiratory distress syndrome and PF [12,13,14,15,16]. HFD could induce various potentially harmful effects to the lungs, including the decreased number and function of mitochondria and increased lung inflammation and abnormal epithelial stem cell proliferation [17]. Observational studies found a significant link between HFD and PF [14,18] as manifested by high intake of SFAs and meat increased the risk of PF and mice lung fibrosis [19,20,21,22]. The exposure to lipids rich diets worsen the airway responsiveness to challenging agent [22,23] and further increased the incidence of PF [24,25]. The involvement of dietary lipids in PF was further supported by the fact that alterations in lipid metabolism enzymes could exacerbate PF progression [26] and that PF patients presented decreased mitochondrial β-oxidation capacity [27]. Taken together these studies reinforced the notion that dietary lipids are direct causative factors of PF which indicated that HFD was tightly associated with the initiation and progression of PF. Nevertheless, the mechanisms that individuals with HFD are susceptible to PF remained unclear. Since genomic mutations induced by HFD are very low [28], it is highly probable that epigenetic changes might contribute to HFD related PF.

Epigenetics is defined as heritable changes in gene expression that are influenced by endogenous and exogenous factors without alterations in the nucleotide sequence [29,30]. The main epigenetic mechanisms included DNA methylation, histone modifications, and non-coding RNAs (ncRNAs) such as microRNAs (miRNAs) and RNA modification [31,32] and dysregulated epigenetic modifications are well recognized player in the development of PF [33,34,35,36,37,38].

In the current review we discussed the effects of HFD induced epigenetic dysregulation on the progression of PF such as epithelial cell injury, abnormal fibroblast activation and chronic inflammation and discussed the possible intervention methods to reduce the risk of HFD related PF. Therefore, this review not only unveiled novel mechanisms but also provided with possible intervention methods to improve the condition of HFD related PF.

## 2. Epithelial Cell Injury and Abnormal Activation

Lung epithelium, principally composed of ciliated cells, basal stem cells and alveolar epithelial cells (AECs), forms a continuous layer and is responsible for gas transportation and exchange, however it is constantly exposed to external insults especially repeated stimulations which would destroy the its integrity [39,40]. Under normal conditions, the damaged lung epithelium repairs itself through activation of local airway or alveolar stem cells (Type 2 alveolar epithelial cells: AEC2s), however dysfunction of these stem cells will hamper this process [41] and apoptotic AEC2s is predominant in idiopathic pulmonary fibrosis (IPF) patients [42]. A recent study showed that HFD slowed resolution of lung fibrosis and delayed alveolar repair by compromising the contribution of AEC2s [20]. Therefore, repeated lung epithelial cell damage and repair dysfunction are widely accepted as the prominent initiation factors of lung fibrosis.

Excessive lipid accumulation or defective fatty acid oxidation is associated with the development of fibrosis [43] and intake of lipids rich diets could trigger the occurrence of PF [24,25]. A recent randomized controlled trial demonstrated that low-carbohydrate HFD could increase serum cholesterol [44] and high cholesterol and HFD could directly induce lipid accumulation in AEC2s thereby up-regulating the expression of Toll like receptor 4 (TLR4) [45] which would lead to apoptosis of AECs and lung fibrosis [46]. However, the possible mechanisms remained unknown. DNA methylation is the main epigenetic mechanism of selective gene expression regulation and methylation on the cytosine residues in CpG dinucleotides representing the simplest form [47]. In mammals, approximately 70% of gene promoter regions contain CpG islands and most which are unmethylated. Traditional concept hold that hypermethylation of CpG islands in gene promoters resulted in gene silencing while hypomethylation lead to activated transcription [47]. While in obese individuals significant lower methylation of CpGs in the first exon of the *TLR4* were observed, therefore these evidence indicated that HFD mediated promoter DNA demethylation contributed to the up-regulation of TLR4 which promoted the apoptosis of AECs and lung fibrosis [46,48]. In addition to AECs apoptosis, several recent studies have unveiled a discrete population of progenitor cells in a “transitional” state that expand dramatically after lung injury [40,49]. These cells, called damage-associated transient progenitor (DATP) cells, are derived from both airway and alveolar epithelial cells and represent a transitional state between injured epithelial cells and newly regenerated alveoli [50]. DATPs are featured by pro-fibrotic phenotype and could facilitate lung fibrosis progression while DNA damage response (DDR) pathway is known to be critical to the DATP phenotype [50]. Accordingly recent studies high light the critical role of increased lung epithelium DDR in the process of PF [50,51]. Histones are vital proteins that compress DNA within the nucleus to form chromatin therefore provide a platform for gene transcription regulation through variety kinds of post-translational modifications such as acetylation, methylation, ubiquitylation and phosphorylation [52]. Specifically histone acetylation and methylation are the most common post-translational modifications that occur at specific sites and residues, which subsequently control gene expression through regulating DNA accessibility to transcription factors and RNA polymerase II [53]. Histone acetylation diminishes the interaction between histones and DNA due to the neutralization of the lysine residue which open the chromatin structure therefore hyperacetylation resulted in gene activation [52]. A recent study showed that HFD could exacerbate particulate matters induced DDR and lung injury by enhancing histone H4 acetylation at lysine 12 (H4K12Ac) as manifested by up-regulation of DNA damage marker molecules such as the total p53, phosphorylated -p53, total Chk1, phosphorylated -Chk1 and γ-H2AX [54]. Taken together the above evidence indicated that HFD could augment the DDR in lung through promoting histone acetylation.

Epithelial mesenchymal transition (EMT), which is featured by loss of epithelial cell-cell adhesion markers, such as E-cadherin and gain of mesenchymal molecules, such as N-cadherin, Vimentin and α-SMA is an initial step in the process of physiological wound healing [55,56]. However, dysregulated EMT contributed to fibrotic diseases progression [57,58]. The role of HFD in EMT is emerging. A recent study showed that HFD promoted EMT in lung tissue as manifested by down regulation of E-cadherin and increased expression of Twist contributed to the formation of lung airway fibrosis but the potential mechanisms remained unclear [59]. Histone methylation generally occurs at arginine residues (such as H3R2 and H3R8) and lysine residues (such as H3K4 and H3K9) [60]. However, unlike histone acetylation, the regulation effects of histone methylation depend on the methylation site. For example, methylation of H3K4, H3R17 and H3K36 were found in transcriptionally active regions, whereas methylation of H3K9, H3K27 and H4K20 were found in transcriptionally repressed regions [61]. Recent studies showed that HFD could significantly increase the histone H3 trimethylation at lysine 4 (H3K4Me3) [62] while increased H3K4Me3 level at the interleukin 6 (*Il-6*) promoter region could promote its expression which subsequently contributed to lung fibrosis through enhancing EMT [63]. Furthermore HFD could increase histone H4 acetylation at lysine 8 (H4K8Ac) and H4K12Ac modification [54] which accounted for EMT in human bronchial epithelial cells and lung fibrosis progression [57,64]. This evidence suggested that histone modification is crucial for HFD mediated EMT in lung fibrosis progression.

Fibrotic diseases are typically characterized by up-regulation of transforming growth factor-beta (TGF-β) and its expression was often correlated with disease severity [65]. TGF-β is a dimeric polypeptide growth factor that regulates cell proliferation and differentiation. In the wound healing process upon epithelium injury, TGF-β could promote the production of collagen, fibronectin and proteoglycans and reduce the function of enzymes involved in ECM degradation such as collagenase and matrix metalloproteinases, thereby resulting in accumulated ECM deposition [53].

A previous study found that HFD could augment the expression of TGF-β which was accompanied by ECM deposition and pro-fibrotic factors expression and TGF-β was mainly originated from in epithelial cells rather than inflammatory cells, suggesting the direct effect of HFD on lung epithelial cell [24]. To promote fibrotic remodeling, latent TGF-β must be converted to active form [66]. A recent study showed that HFD could increase the transformation of latent TGF-β into active state in bronchial epithelium, thereby facilitating lung fibrosis [1,22] and this transformation was partially controlled by epigenetic regulation since increasing DNA methylation (5-mC) levels in HFD mice could decrease TGF-β expression [67]. Similarly, in HFD treated rats, the n6-methyladenosine (m6A) modification on the 5′ untranslated region (UTR) of TGF-β mRNA was increased which resulted in m6A-dependent translation of TGF-β mRNA [68] and in HFD induced kidney fibrosis, reduced occupancy of histone H2A lysine 119 mono-ubiquitination (H2AK119Ub) at promoter region facilitated the expression of TGF-β [69]. Furthermore HFD could also decrease miRNAs such as let7, mir-21 and mir-27 to activate TGF-β signaling by promoting TGFβRIII expression [70]. These studies strongly supported the notion that epigenetic regulation was vital for HFD induced TGF-β signaling activation.

Upon lung injury, type 1 alveolar epithelial cells (AEC1s) are susceptible to injury while AEC2s serve as progenitor cells which could proliferate and differentiate to create new AEC2s and AEC1s [71]. Altered response of the lung epithelial stem cells (AEC2s and club cells) to injuries is known to be the major contributor to lung fibrosis [72]. These stem cells might over proliferate and cause aberrant repair and therefore forming fibroblast foci [72,73]. A recent study showed that HFD slowed resolution of lung fibrosis and delays alveolar repair by compromising the contribution of AEC2s [20] which was due to impairment of mitochondrial function of AEC2s induced by HFD [17]. Fatty acid synthase (FASN) is the lone lipogenic enzyme that was able to synthesize fatty acids de novo in humans [74], while a recent study showed that HFD could decrease FASN expression in AEC2s which resulted in mitochondrial dysfunction and more severe lung injury [75]. Physiologically the dysfunctional mitochondria were eliminated by PINK1/Parkin-mediated mitophagy [76] however, HFD could compromise mitochondrial homeostasis through increasing histone H3 acetylation at lysine 27 (H3K27Ac) at the promoter of PINK1 which subsequently inhibited its expression, thereby deactivating PINK1/Parkin-mediated mitophagy [77]. Therefore, deficient mitophagy caused by HFD could compromise the mitochondrial function in AEC2s, thereby facilitating fibrosis progression.

Endoplasmic reticulum (ER) stress refers to accumulation of unfolded proteins induced by imbalanced protein homeostasis [78]. Excessive ER stress lead to repetitive damages to epithelial cells and was recently showed to facilitate PF progression [1,79]. Diets with a high-fat content are associated with elevated circulating free fatty acids, which could induce ER stress in non-adipose tissues [80]. A recent study demonstrated that administration of HFD in mice could increase epithelial ER stress and susceptibility to lung fibrosis [21] while ER stress could be evoked by HFD through decreasing methylated histones in *Lxrα* promoter [28]. Taken together the above evidence suggested that epigenetic regulation played significant roles in HFD mediated lung epithelial cell injury and aberrant activation which facilitated lung fibrosis progression (Figure 1).

## 3. Uncontrolled Fibroblast Activation

Due to its tremendous surface area, the lung epithelium is constantly faced with external insults while the repair process not only required the activation of lung epithelium stem cells but also fibroblast to secrete ECM which generated contractile force for repairing the damaged cells and wound closure [39,40]. Ideally this process is temporary and normal pulmonary homeostasis will be restored, however under pathologically conditions this process was sustained and massive ECM were generated, thereby leading to fibrosis [1,81]. Fibroblasts from PF patients exhibited an abnormally “activated” phenotype and alterations in DNA methylation contribute to activation of fibroblasts [81]. Consistently, mounting evidence indicated that abnormal activation of lung fibroblasts played a vital role in modulating repair after lung injury, thereby contributing to fibrotic progression [1,82]. However, the possible roles of fibroblast played in HFD induced PF were still unclear. A recent study showed that high-fat and high-fructose diet could facilitate lung fibrosis through inducing lung fibroblasts inflammation via transcriptional up-regulating dedicator of cytokinesis 2 (DOCK2) [83]. Accordingly, miR-34 was crucial for HFD induced fibrotic genes expression such as Col6a1, MMP12, and TGF-β [84].

Cholesterol from HFD was one of the major bioactive lipid molecules that were dysregulated in many chronic lung diseases such as chronic obstructive pulmonary disease and lung fibrosis [85,86]. The link between cholesterol and fibrosis is becoming progressively clearer. The development of HFD induced hypercholesterolemia in *ApoE*^−/−^ mice could contribute to lipid accumulation within the lung and subsequent fibrosis [87] while Atorvastatin could attenuate PF in mice by regulating myofibroblast differentiation and apoptosis [88]. Under physiological conditions the lung fibroblasts secrete little ECM, however, under fibrotic conditions over proliferated fibroblast, which was featured by expression of proliferative phenotype marker Cyclin D1 could secrete more ECM, therefore, inhibiting fibroblast proliferation has become an important treatment strategy for PF [89]. Mechanistically, cholesterol could significantly recover the reduced chromatin accessibility induced by Pitavastatin through restoring H3K27Ac at the enhancer regions of *Myc* [90]. Accordingly HFD could enhance *Myc* transcriptional through enhancing histone H4K20 hypomethylation at the promoter regions [91] while MYC further promoted proliferation and differentiation of lung fibroblasts, thereby facilitating the progression of PF [92]. Furthermore, HFD feeding could give rise to type I collagen-depositing fibroblasts [93]. The above evidence indicated that cholesterol lowing drug could be the potential drugs for HFD related lung fibrosis.

A previous clinical study observed a positive correlation between SFAs intake and increased risk of IPF in a Japanese cohort which suggested that high SFAs intake could increase the risk of IPF [13] and a recent study demonstrated that HFD rich in palmitic acid (PA) (a kind of SFA) could promote PF in mice [21]. These results were in accordance with the fact that elevated relative PA contents were observed in IPF patients [94]. Furthermore SFAs such as PA could generate a metabolic memory by increasing histone H3 dimethylation at lysine 36 (H3K36Me2) and decreasing histone H3 trimethylation at lysine 27 (H3K27Me3) on *Foxo1* promoter region to increase its activity [95] which was crucial for activated fibroblast to secret ECM in the progression of lung fibrosis [96].

Once the wound healing for lung epithelium repair is complete, the unrequired proliferated and activated fibroblasts should be eliminated through apoptosis to limit excess ECM deposition while impaired apoptosis of activated fibroblasts could lead to tissue fibrosis. Indeed, fibrotic lung fibroblasts seem to be resistant to apoptosis [71]. However, the factors that distinguish normal wound repairs from fibrosis remained unknown. PA in HFD could promote histone H3 trimethylation at lysine 9 (H3K9Me3) on the promoter region [97] which resulted in the decreased expression of the death receptor Fas, thereby facilitating the resistance of fibroblasts to Fas-mediated apoptosis in fibrotic lung [98]. Furthermore recent studies showed that HFD induced obesity through promoting histone H4 acetylation at lysine 16 (H4K16Ac) [99] which could modulate chromatin structure by serving as a switch from a repressive to a transcriptionally active state and promote the expression of pro-fibrotic genes to facilitate fibroblast collagen deposition in lung fibroblasts from IPF patients and mice lung fibrosis model [100]. Thus, HFD mediated H4K16Ac modification in obese population could promote lung fibrosis development via accelerating fibroblast collagen deposition.

Circular RNAs (circRNAs), covalently closed at the 5′ and 3′ end, are novel ncRNAs that regulating gene expression mainly though interaction with miRNAs or proteins, regulation of transcription and translation into peptides [101]. Recent study uncovered the critical roles of CircRNAs as important epigenetic regulator in the pathogenesis of PF [101]. A recent study demonstrated that lipid accumulation caused by HFD could decrease mitochondrial circRNA Steatohepatitis-associated circRNA ATP5B Regulator (SCAR) which could inhibit fibroblast activation by decreasing the expression of collagen and α-SMA, while nanoparticle delivering circRNA SCAR could suppress fibroblast activation in HFD treated mice [102]. Although this observation is in liver, the development of organ fibrosis shared common pathways and this could provide us with potential future direction especially on drugs targeting epigenetic regulation [103,104,105].

Taken together, dysregulated epigenetic modifications are crucial for HFD induced fibroblast activation, proliferation and apoptosis resistance, all of which are dominant contributors to the progression of lung fibrosis (Figure 2).

## 4. Chronic Inflammation

Chronic inflammation played important roles in the progression of PF [106]. In addition to the above mentioned repair mechanisms: activation of lung epithelium stem cells and fibroblast upon injury, the immune response was simultaneously triggered to protect the tissue from further damages [107]. However, this process is out of control with chronic repeated injuries (typically observed in lung fibrosis) [108,109]. Due to continuous inflammation, secretion of pro-inflammatory cytokines such as IL-6 and tumor necrosis factor α (TNFα) will be increased which could further augment wound healing process by facilitating matrix deposition and subsequently resulted in fibrotic progression [103,104]. Therefore identification of factors participating in the onset and progression of inflammation is vital for comprehensively understanding inflammation related disorders such as PF [106,110].

Nutritional factors contributed to the formation of pro-inflammatory niche [111] and chronic low-grade inflammation induced by altered metabolic homeostasis appeared to be vital for the pathogenesis of organ fibrosis [112]. Chronic HFD has been linked with low-grade systemic inflammation in obesity [113]. Particularly, the consumption of western type HFD could provoke chronic metabolic inflammation which subsequently contributed to the progression of chronic diseases such as nonalcoholic steatohepatitis and lung fibrosis [114]. In HFD related obesity, adipocytes and macrophages in the adipose tissue generated pro-inflammatory cytokines such as TNFα and IL-6 which could provoke systemic inflammation and contribute to the progression of lung fibrosis [115,116]. A recent report showed that mice displayed increased pulmonary neutrophile accumulation and collagen deposition by feeding with HFD [25] and mice feed on HFD exhibited granulomatous lung inflammations which subsequently lead to progressive lung fibrosis [117]. However, the mechanisms through which HFD provoked inflammation response remained unclear. It has been well established that epigenetic modifications upon environmental factor stimulation played a fundamental role in regulation of inflammatory gene transcription [110,118,119,120]. Worse still, the adverse effects of the inflammatory state may induce epigenetic changes that perpetuate inflammation [121]. Therefore, we postulated that epigenetic signature alterations induced by HFD may exacerbate inflammatory responses, thereby influencing progression of chronic inflammatory disease such as lung fibrosis [101,110,118]. In support of this, integrative epigenome wide association study showed that promoter methylation of *TNFA* were decreased with consumption of dietary fat [122] suggesting a nutrient epigenomic regulation of pro-inflammatory factors [123]. Accordingly high cholesterol and HFD would lead to low-grade pulmonary inflammation through activating TLR4/NFκB signaling [45] while significant lower methylation of CpGs in the first exon of the *TLR4* were observed in obese individuals, indicating epigenetic regulation of TLR4 expression in obesity [48]. Mice fed on HFD exhibited significant reduced DNA methylation at the promoter of *Pparγ1* which was critical for pro-inflammatory macrophages activation [124]. In adipose tissue, the DNMT3a methyltransferase was markedly increased which was accompanied by elevated expression of inflammatory cytokines such as TNFα and MCP-1, implying the role of DNMT3a in obesity related inflammation [125]. Similarly, increased expression of DNMT3b was found in adipose tissue macrophages and involved in the polarization of macrophage and inflammation [126]. In mice, HFD led to hypermethylation of the *Ankrd26* which in turn contributed to enhanced secretion of pro-inflammatory factors [127]. Consistently, epigenetic silencing of the *ANKRD26* by promoter methylation was related to pro-inflammatory state in obese individuals [128]. In addition to DNA methylation, significant association between expression of histone deacetylases and inflammation status was demonstrated in obese individuals [129]. Upon HFD treatment the levels of sphingosine-1-phosphate were increased [130] which subsequently inhibited histone deacetylases activity and increased histone acetylation at H3K9, H4K8 and H3K18, thereby promoting pro-inflammatory cytokines in BALF [131]. Accordingly, sphingosine-1-phosphate was reported to be increased in IPF patients and could facilitate disease progression [132,133]. Moreover ncRNAs, due to their versatile roles in the regulation of gene expression, are widely involved in HFD induced chronic inflammation. Consumption of high-fat or high calorie (rich in fat) diet was shown to increase inflammatory response by altering miRNA expression [134,135] and bioinformatics study showed that a miRNAs network significantly associated with obesity related inflammation [136] and deregulated circulating inflammatory miRNAs contributed to the elevated inflammatory state in obesity [137]. Besides, adipocyte-secreted exosomal miR-34 was progressively increased with the development of dietary obesity and subsequent systemic inflammation [138]. In the same way, HFD could increase miR-155 in adipocyte-derived microvesicles which could induce M1 macrophage polarization, thereby causing chronic inflammation [139]. HFD could further down regulate miR-30 by DNA methylation which facilitated M1 macrophages polarization [140]. Although the pro-inflammatory M1 macrophages are usually regarded as anti-fibrotic in lung fibrosis, they could exacerbate the inflammatory status of the lung injury and evoke the fibrotic response in lung fibrosis patients through activation of the TLR4 signaling [141]. Worse still, the maternal HFD could further hinder the lung development and function of offspring by epigenetic modulations [142,143] for example, maternal HFD could lead to offspring tissue inflammation through down regulation of miR-706 [144]. Since HFD could also suppress the expression of miR-26a and stimulate expression of pro-inflammatory cytokines such as TNFα [145] while decreasing TNFα was demonstrated to improve lung function of PF patient [146] therefore, this provide us with novel target for treating HFD related PF. Taken together the above evidence highlighted the crucial roles of epigenetic regulated inflammation in HFD induced lung fibrosis (Figure 3).

## 5. Clinical Perspectives

In the current review we emphasized the important roles of epigenetic regulation in HFD related lung fibrosis. HFD is well accepted as a critical factor leading to the obesity [11]. Paradoxically, previous multicenter study showed that body weight loss predicted worse survival of PF patients [147]. This issue was due to the currently used body weight measurement which neglected the body mass composition [148]. Actually individuals with the same body mass may varied in composition including fat mass (FM) and fat-free mass (FFM; or lean mass) which played different roles in health outcomes [148,149]. Large prospective cohort studies demonstrated that increased FM could significantly increase the risk while FFM reduced the risk of inflammation related and respiratory diseases [149,150] which mean that hidden loss of FFM or lean mass rather than weight loss was related to increased systemic inflammatory [151] since elevated expression of inflammation related genes were induced by HFD related FM increase [152]. On the contrary increased proportion of FFM was associated with better lung condition [153] which could be attributed to lower inflammation.

It is well established that high intake of polyunsaturated fatty acids (PUFAs) has been associated with reduction of adiposity and increases in lean body mass [154]. However in the last decades, the daily diets FA intake has dramatically changed from monounsaturated and PUFAs rich pattern to a westernized pattern characterized by a high content in SFAs [155]. Accordingly, a previous comparative study showed that SFAs intake could increase the risk of PF [13] while the beneficial effects of PUFAs on mitigating lung fibrosis have been demonstrated in many studies. Intake of fish oil rich in eicosapentaenoic acid decreased bleomycin induced lung hydroxyproline accumulation [156]. Furthermore, the mitigation of lung fibrosis has been demonstrated with long-chain ω-3 PUFA docosahexaenoic acid [157] and short-chain ω-3 PUFA [158,159]. A relevant case showed that maternal diet supplied with docosahexaenoic acid could alleviate lung fibrosis and improve lung function in offspring by reducing collagen deposition and lessening inflammation [160]. These anti-fibrotic properties of PUFA could be mediated through inhibiting EMT in human AEC2s [161] and through activating PPARγ signaling [162]. Since HFD is critical contributor to fat body mass increase and obesity [11,152] while intake of PUFAs has been associated with reduction of adiposity and increases in fat-free body mass [154] therefore, an adequate dietary PUFAs intake might reduce the risk of HFD related lung fibrosis. Indeed observational data from a cohort of 104 Japanese patients showed that SFAs intake may be an independent risk factor for PF [13] while consumption of fruit was associated with a reduced risk [163]. Therefore a shift of dietary habit should be recommended for individuals with a high fat mass to avoid the occurrence of PF. Alternatively, an relative easy way for HFD individual to reduce the risk of lung fibrosis might be exercise since a latest study demonstrated that aerobic exercise could alleviate PF by ameliorating HFD induced inflammatory response and neutrophil infiltration [164].

## 6. Conclusions

Lung fibrosis is an interstitial lung disease characterized by chronic inflammation and destruction of lung parenchyma which was caused by accelerated ECM deposition [1]. Upon injury, normal stem cell activation and wound healing procedure will lead to epithelium repair while abnormal lung epithelium cell, fibroblast activation and accompanied chronic inflammation will result in tissue fibrosis which impaired gas exchange and lead to breathlessness, thereby decreasing quality of life. However, the mechanisms that shift normal repair to fibrotic response remained unclear.

Recently, lung fibrosis is recognized as a metabolic disease and abnormal lipid signature was observed both in serum and BALF of PF patients and mice model, suggesting that lipid metabolism was unbalanced in lung fibrosis. Consistently clinical observation and animal studies showed that HFD was associated with the progression of lung fibrosis [14,18,21,24,25]. However, the mechanisms of individuals with HFD are susceptible to lung fibrosis remained unclear. Since genomic mutation induced by HFD is very low, it is highly probable that epigenetic changes might contribute to HFD related lung fibrosis. In the current review we highlight the vital roles of epigenetic dysregulation in HFD induced PF from the perspective of epithelial cell injury, abnormal fibroblast activation and chronic inflammation. This knowledge opens new possibilities for a potential use of epigenetic signatures as biomarkers for diagnosis and targets for PF management [110]. Currently, there is no cure for PF except for lung transplantation therefore, revealing the potential pathogenic factors and possible mechanisms would contribute to the prevention and treatment of this deadly disease. Due to the reversible nature, intervention methods targeting dysregulated epigenetic regulation represented a promising way to treat lung fibrosis [165,166,167]. For a long time we have studied on the therapeutic effects of miRNAs mimics in treating lung fibrosis [168,169]. Recently, we generated MRG-229, a next-generation miR-29 mimic with improved stability and potential for targeted delivery which showed significant anti-fibrotic effects on human precision cut lung slices and mice lung fibrosis model and showed no adverse effects on non-human primates cynomolgus monkeys [168]. Accordingly, delivering circRNA SCAR using nanoparticle could suppress fibroblast activation in HFD treated mice [102]. The above evidence demonstrated the vital roles of targeting abnormal epigenetic regulation in ameliorating PF progression.

In summary, our review not only unveil the important roles of epigenetic regulation in HFD mediated PF but also provide potential ways to deal with this issue. For patients they could change their diet habitat and do more aerobic exercise [158,159,164] while for scientific researchers or drug developers, unveiling the epigenetic mechanism of HFD related lung fibrosis will provide novel targets to treat this deadly disease.

## Figures and Tables

**Figure 1 biomolecules-13-00558-f001:**
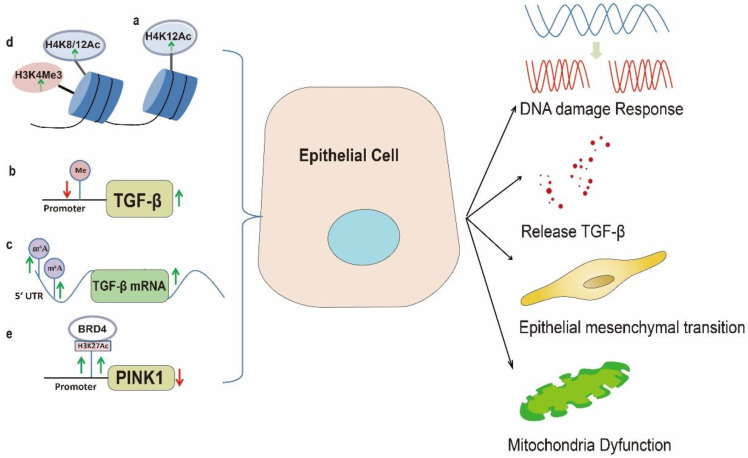
Epigenetic dysregulation induced by HFD contributed to lung fibrosis through epithelium injury and abnormal activation. (**a**): HFD exacerbates DNA damage response and lung epithelium injury by enhancing H4K12Ac; (**b**): HFD decreases promoter DNA methylation levels in mice whereby increasing pro-fibrotic molecule TGF-β expression; (**c**): In HFD treated rats the m6A modification on the 5′UTR of TGF-β mRNA is increased which resulted in m6A-dependent translation of TGF-β mRNA; (**d**): HFD could significantly increase the H3K4Me3, H4K8Ac and H4K12Ac modification which are accounted for EMT in human bronchial epithelial cells, thereby contributing to lung fibrosis; (**e**): HFD could compromise mitochondrial homeostasis through increasing H3K27Ac at the promoter of PINK1 which subsequently inhibits its expression, thereby inducing mitochondria dysfunction.

**Figure 2 biomolecules-13-00558-f002:**
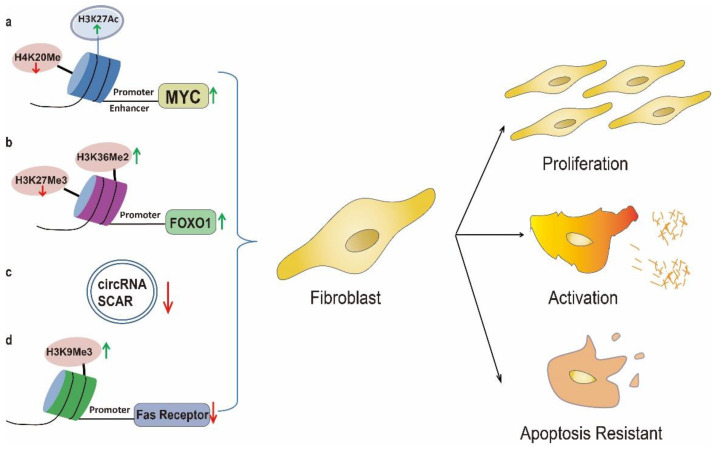
HFD facilitates the proliferation, over activation and apoptosis resistant through epigenetic modification. (**a**): HFD could enhance MYC transcriptional through enhancing H4K20 hypomethylation and H3K27Ac at the promoter regions which promotes lung fibroblast proliferation; (**b**): HFD rich in PA could activate *Foxo1* through increasing H3K36Me2 and decreasing H3K27Me3 on promoter region which is crucial for activated fibroblast to secret collagen in the progression of lung fibrosis; (**c**): HFD could decrease mitochondrial circRNA SCAR to facilitate fibroblast activation; (**d**): PA could promote H3K9Me3 which is responsible for the decreased expression of the death receptor Fas thus facilitates fibroblasts resistance to apoptosis in fibrotic lung.

**Figure 3 biomolecules-13-00558-f003:**
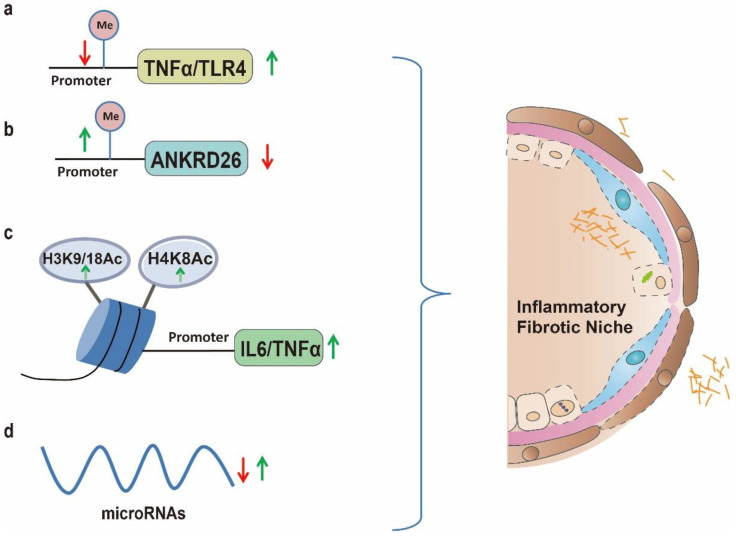
Epigenetically activation of inflammatory related genes play significant roles in fostering inflammation niche by HFD, thereby facilitating lung fibrosis progression. (**a**): HFD promotes TLR4 expression through demethylation of CpGs in the first exon whereby lead to low-grade pulmonary inflammation; (**b**): HFD led to hypermethylation of the *Ankrd26* promoter region which in turn contribute to enhanced secretion of pro-inflammatory factors; (**c**): HFD increases histone acetylation at H3K9, H4K8, H3K18 which promoted pro-inflammatory cytokines in BALF; (**d**): Consumption of high-fat or high calorie diet (rich in fat) is shown to increase inflammatory response by altering miRNA expression such as increasing miR-155 and down regulating miR-30 could induce M1 macrophage polarization, thereby causing chronic inflammation.

## Data Availability

No new data were generated in this study therefore data sharing is not applicable to this article.

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
