# Peer review of "High-Fat Diet Related Lung Fibrosis-Epigenetic Regulation Matters"

_biomolecules, 2023, doi:10.3390/biom13030558_

Round 1
Reviewer 1 Report
The manuscript titled "High fat diet related lung fibrosis-epigenetic regulation matters" is a systemic review discussing the role of epigenetic dysregulation induced due to high fat diet on lung fibrosis. The reviewer has following comments:
(i) The authors should be cautious of the abbreviation that they use in the text only once or a few times. For instance, PM for particulate matter is just used once. Other instance is the use of of abbreviation such as "RSV" for resveratrol. However, RSV is commonly used for respiratory syncytial virus. Use of RSV in the text deviates the attention to virus with respect to the context of the study.
(ii) In line 170, authors discuss the role of protein FASN (??) and in line 169, AT2. The authors should briefly discuss what these proteins are in context of physiological function of mitochondria.
(iii) In figure 1, do authors talk about mitochondrial dysfunction of epithelial cells induced by HFD, because in the lines 169-170, authors talk about HFD induced impairment of mitochondrial function of lung stem cells. Authors should comment on this discrepancy.
(iv) In lines 233, authors should clarify what they mean by SFA
(v) Authors should be extremely cautious of spellings which can completely change the meaning of a word; in line 330, is it "non" or "none" coding RNAs?
(vi) Authors should clarify if high fat/ high calorie diet (lines 338 and 365) mean the same.
Reviewer 2 Report
Please review the attached file.

Reviewer 3 Report
In this manuscript, Juntang Yang et al. describe in a very elegant way the effects of the epigenetic dysregulation induced by HFD on fibrotic factors such as epithelial cell injury, abnormal fibroblast activation, and chronic inflammation.
This is a growing area and a review focused on epigenetic changes will surely be of great interest to the readers.
References are appropriate, most of them are up-to-date and refer to what has been discussed.
The manuscript is very well organized and written, has its merits, and is relevant to scientific interest. However, I would like to make some suggestions to improve the manuscript
1. The conclusions are adequate; however, I believe it lacks the essence of the review.
2. Do not define in order of appearance some abbreviations, unify terms such as stem or progenitor cells, m6A, AT2, and FASN.
3 In lines 157a 159 define in which tissue it is (reference 59).
4 Mystypo in image 1 dyfunction, change dysfunction
5 Make images more didactic
6 Add other HFD-related circRNAs are suggested.
suggestion if there are enough references;
It would be interesting to add an exclusive section for myofibroblasts or macrophages in HFD.
Round 2
Reviewer 2 Report
COMMENTS TO THE AUTHORS:
In this review article the authors approach pulmonary fibrosis from the metabolic concept, in particular, the effect of high-fat diet (HFD) on epigenetic modifications that impact relevant processes in pulmonary fibrogenesis.
The authors made the modifications requested in the previous review, however, some additional minor corrections are noted below.
§ Define all abbreviations at their first occurrence and do not redefine in the text, e.g. page 4, line 167.
§ Correctly use et al. when referencing or, where appropriate, find the correct abbreviation for etcetera.
§ Write progression correctly, page 3, line 135.
§ Abbreviations section homogenise formatting, page 11, line 460.
